# Oral Squamous Cell Carcinoma Contributes to Differentiation of Monocyte-Derived Tumor-Associated Macrophages via PAI-1 and IL-8 Production

**DOI:** 10.3390/ijms22179475

**Published:** 2021-08-31

**Authors:** Kazuki Kai, Masafumi Moriyama, A. S. M. Rafiul Haque, Taichi Hattori, Akira Chinju, Chen Hu, Keigo Kubota, Yuka Miyahara, Noriko Kakizoe-Ishiguro, Shintaro Kawano, Seiji Nakamura

**Affiliations:** 1Section of Oral and Maxillofacial Oncology, Division of Maxillofacial Diagnostic and Surgical Sciences, Faculty of Dental Science, Kyushu University, 3-1-1 Maidashi, Higashi-ku, Fukuoka 812-8582, Japan; k.kai@dent.kyushu-u.ac.jp (K.K.); rafi07.dr@gmail.com (A.S.M.R.H.); hattori91@dent.kyushu-u.ac.jp (T.H.); chinju@dent.kyushu-u.ac.jp (A.C.); chenhu96lkk@gmail.com (C.H.); yuka.m0119@dent.kyushu-u.ac.jp (Y.M.); ishiguro@dent.kyushu-u.ac.jp (N.K.-I.); skawano@dent.kyushu-u.ac.jp (S.K.); seiji@dent.kyushu-u.ac.jp (S.N.); 2OBT Research Center, Faculty of Dental Science, Kyushu University, 3-1-1 Maidashi, Higashi-ku, Fukuoka 812-8582, Japan; 3Department of Dental Anatomy, Udayan Dental College, House No: 1, Ward No: 7, Chondipur, GPO, Rajpara, Rajshahi 6000, Bangladesh; 4Department of Oral-Maxillofacial Surgery, Dentistry and Orthodontics, The University of Tokyo Hospital, 7-3-1 Hongo, Bunkyo-ku, Tokyo 113-8655, Japan; kubotak.ora@gmail.com

**Keywords:** tumor-associated macrophage, oral squamous cell carcinoma, CD206, plasminogen activator inhibitor–1, interleukin-8

## Abstract

Tumor-associated macrophages (TAMs) promote cancer cell proliferation and metastasis, as well as anti-tumor immune suppression. Recent studies have shown that tumors enhance the recruitment and differentiation of TAMs, but the detailed mechanisms have not been clarified. We thus examined the influence of cancer cells on the differentiation of monocytes to TAM subsets, including CD163^+^, CD204^+^, and CD206^+^ cells, in oral squamous cell carcinoma (OSCC) using immunohistochemistry, flow cytometry, and a cytokine array. Furthermore, we investigated the effect of OSCC cells (HSC-2, SQUU-A, and SQUU-B cells) on the differentiation of purified CD14^+^ cells to TAM subsets. The localization patterns of CD163^+^, CD204^+^, and CD206^+^ in OSCC sections were quite different. The expression of CD206 on CD14^+^ cells was significantly increased after the co-culture with OSCC cell lines, while the expressions of CD163 and CD204 on CD14^+^ cells showed no change. High concentrations of plasminogen activator inhibitor-1 (PAI-1) and interleukin-8 (IL-8) were detected in the conditioned medium of OSCC cell lines. PAI-1 and IL-8 stimulated CD14^+^ cells to express CD206. Moreover, there were positive correlations among the numbers of CD206^+^, PAI-1^+^, and IL-8^+^ cells in OSCC sections. These results suggest that PAI-1 and IL-8 produced by OSCC contribute to the differentiation of monocytes to CD206^+^ TAMs.

## 1. Introduction

Oral squamous cell carcinoma (OSCC) represents approximately 90% of all malignant neoplasms of the head and neck and is associated with a high mortality rate. The development of diagnostic tests, surgical techniques, radiotherapy, chemotherapeutic drugs, and other supportive treatments has improved the diagnosis and prognosis of early-stage OSCC. However, more than 400,000 people are affected by OSCC per year worldwide, and the five-year survival rate of advanced-stage OSCC has remained low for decades because of its rapid progression and high metastasis and recurrence capability [1,2,3,4,5]. Therefore, a better understanding of the mechanisms underlying the initiation, progression, and metastasis of OSCC is required to identify potential treatments to improve the poor outcome of this disease.

The activation of macrophages is the main event in the pathogenesis of multiple diseases. Macrophages are generally classified into two distinct subtypes: classically activated M1 macrophages, which are stimulated by interferon (IFN)-γ or lipopolysaccharides; and alternatively activated M2 macrophages, which are stimulated by T helper-2 cytokines such as interleukin (IL)-4, IL-10, or IL-13 [6,7,8,9]. M1 macrophages secrete proinflammatory cytokines and phagocytize microbes, contributing to microbicidal and anti-tumor immunity, while M2 macrophages secrete anti-inflammatory factors and scavenge debris, contributing to angiogenesis, suppression of adaptive immunity, wound healing, and tissue repair. 

In the tumor microenvironment (TME), infiltrating macrophages typically exhibit the M2 phenotype and are called tumor-associated macrophages (TAMs). TAMs are considered the most potent factor that contribute toward tumor cell proliferation, metastasis, and angiogenesis [10]. Furthermore, tumors secrete exogenous factors that enhance the chemotaxis/migration of monocytes and the differentiation of monocytes to TAMs in the TME [11]. However, how tumor cells induce the differentiation of monocytes to TAMs remains unknown.

In this study, we investigated the effect of OSCC tumor cell-soluble factors on the differentiation of monocytes to TAMs in vitro and identified the distinct soluble factors that influence monocyte differentiation. 

## 2. Results 

### 2.1. Distribution of TAM Markers in OSCC Tissues

To examine the localization of the TAM markers CD163, CD204, and CD206 in OSCC sections, immunohistochemical staining was performed. Representative findings are shown in Figure 1A. The expression of CD163 and CD206 was strongly detected in and around tumors and tumor stroma, while the expression of CD204 was mainly detected in and around tumors (Figure 1A). We further performed triple immunofluorescence staining to confirm the distribution of these TAM markers in OSCC tissues, and it was actually different from its distribution (Figure 1B). 

### 2.2. Effects of OSCC Cell Co-Culture on Apoptosis and Activation Status of CD14^+^ Monocytes 

To determine the effect of OSCC on monocytes in vitro, we co-cultured purified CD14^+^ cells with OSCC cell lines (HSC-2, SQUU-A, and SQUU-B cells), as described in the Materials and Methods (Figure 2A), followed by flow cytometric analysis. The number of monocytes after co-culture with the three OSCC cell lines was significantly higher than in the absence of OSCC cells (Figure 2B,C). We further examined the ability of OSCC cell lines to cause apoptosis of purified CD14^+^ cells after co-culture by measuring the levels of 7-AAD^+^ CD14^+^ cells (Appendix A). The number of CD14^+^ cells after co-culture with OSCC cell lines was significantly increased (Figure 2D).

### 2.3. Association of OSCC Cell Lines with Differentiation of Monocytes to TAMs

We next confirmed the effects of OSCC cell lines on the differentiation of purified CD14^+^ cells to TAMs. The expression of CD206 on CD14^+^ cells co-cultured with OSCC cell lines was significantly higher than that in CD14^+^ cells without co-culture, while the expressions of CD163 and CD204 on CD14^+^ cells showed no significant differences with or without co-culture with OSCC cells (Figure 2E,F). 

To further study the mechanism underlying OSCC cells promoting the differentiation of TAMs, a cytokine array was performed to determine the cytokine profiles in the conditioned medium (CM) obtained from OSCC cell lines cultured for 4 days (Figure 3A). Among the detected cytokines, the concentrations of interleukin-8 (IL-8) and plasminogen activator inhibitor-1 (PAI-1) were particularly high (Figure 3B). Moreover, the concentrations of IL-8 and PAI-1 in the CM of OSCC cell lines were significantly higher than those in the CM without OSCC cell lines (CM alone) (Figure 3C). Furthermore, treating CD14^+^ cells with PAI-1 for 4 days led to the increased expression of CD206, and these were highly increased by the addition of IL-8 (Figure 3D,E). We found that 7-AAD^+^ CD14^+^ cells treated with PAI-1 and IL-8 expressed significantly lower levels than those without PAI-1 or IL-8 (Figure 3F).

### 2.4. Distribution of IL-8 and PAI-1 in OSCC Tissues

We next examined the localizations of PAI-1 and IL-8 in the OSCC section. Representative findings are shown in Figure 4A. The expression of PAI-1 and IL-8 was strongly detected in and around tumors, especially tumors with a high grade of malignancy, and it was actually different from its distribution (Figure 4B). In addition, we observed significant positive correlations among the numbers of CD206^+^, PAI-1^+^, and IL-8^+^ cells in 18 OSCC samples (Figure 4C). 

### 2.5. Associations of IL-8 and PAI-1 with Clinical Outcomes and Prognosis of OSCC Patients

To evaluate the involvement of IL-8 and PAI-1 in the clinical prognosis of OSCC patients, survival rates were estimated by using the Kaplan–Meier method. The OSCC patients were divided into low- and high-expression groups according to the mean number of IL-8^+^, PAI-1^+^, or CD206^+^ cells. In the disease-specific survival rate, there were no significant differences in low and high IL-8^+^, PAI-1^+^, or CD206^+^ expression. In the progression-free survival rate, patients with high IL-8^+^ or PAI-1^+^ expression had a significantly more unfavorable outcome than those with low-expression groups. On the other hand, patients with high CD206^+^ expression showed a lower progression-free survival rate than those with the low-CD206^+^-expression group, but there was no significant difference between low- and high-CD206^+^-expression groups (Figure 5). According to the above results, a higher sensitivity and specificity of IL-8^+^ and PAI-1^+^ expression were found for adverse prognosis. 

## 3. Discussion

TAMs are a major component of the TME in OSCC and other solid tumors. Recent studies have demonstrated that TAMs play important roles in tumor progression in OSCC (summarized briefly in Table 1). The number and distribution of infiltrating TAMs are strongly correlated with anti-tumor immunosuppression, tumor invasiveness, distant metastasis, poor clinical outcome, and shortened overall survival. CD163, CD204, and CD206 are considered markers for the activation of TAMs, as well as M2 macrophages. 

CD163 is a member of the scavenger receptor cysteine-rich family class B receptor family and is mainly expressed on mature tissue macrophages [12]. The main function of CD163 is the binding of the hemoglobin–haptoglobin complex. CD163-positive macrophages infiltrate into inflammatory tissues and are involved in the resolution of inflammation [13]. CD163 is mainly expressed on monocyte-macrophages and is a potential diagnostic parameter for monitoring the activity of macrophages in inflammatory disease [14]. 

CD204 is a prototypic member of a family of structurally different transmembrane receptors conjointly called scavenger receptors and is primarily expressed on macrophages and dendritic cells. CD204 recognizes various molecules and factors such as modified lipid proteins, apoptotic cells, and exogenous pathogen-associated molecular patterns, which results in the promotion of lipid metabolism, atherogenesis, and metabolic processes [15,16]. CD204-deficient macrophages exhibit tumor-inhibitory activity through the secretion of nitric oxide and interferons [17], and CD204 has been implicated as a suppressor of the inflammatory response [18]. Our previous study demonstrated that CD163^+^CD204^+^ TAMs promote the invasion and metastasis of OSCC by T-cell regulation via IL-10 and PD-L1 [19].

CD206 is a C-type lectin, also known as the macrophage mannose receptor, which is expressed by tissue macrophages, dendritic cells, and specific lymphatic or endothelial cells. CD206 plays an important role in immune homeostasis, but its high expression has been detected in the TME [20]. CD206 is strongly expressed in prostate adenocarcinoma, and the number of CD206^+^ TAMs strongly correlates with poor prognosis of the disease [21]. Furthermore, Dong et al. showed that a high CD206^+^ TAM density in hepatocellular carcinoma correlated with aggressive tumor phenotypes, including poor tumor differentiation and advanced TNM stage, leading to poor prognosis and recurrence [22].

In the present study, we performed immunohistochemical staining to determine the distribution of the TAM markers (CD163, CD204, and CD206) in OSCC tissues, and it was actually different from its distribution (Figure 1B). As the polarization and activation of TAMs is a complex process that varies across different tumors, identifying common specific TAM subsets in each tumor type may be difficult. 

We recently found that the EGF concentration in the CM of CD206^+^ TAMs from OSCC was significantly higher than the levels in the CM of CD163^+^ and CD204^+^ TAMs. OSCC cell lines (HSC-2 cells) cultured in the CM of CD206^+^ TAMs showed highly increased cell proliferation and invasion activities compared with cells cultured in the CM of CD163^+^ or CD204^+^ TAMs. Notably, anti-EGFR antibodies significantly reduced the viability of HSC-2 cells cultured in the CM of TAM subsets, especially CD206^+^ TAMs. These results suggest that CD206^+^ TAMs promote the proliferation and invasion of OSCC cells via EGF production. A high expression of CD206^+^ TAMs correlated with unfavorable clinical prognosis in OSCC patients [23]. Interestingly, recent studies have indicated that EGFR signaling induced IL-8 and PAI-1 production in pulmonary epithelial cells or keratinocytes [24,25].

In the present study, our findings showed that OSCC cells promote the differentiation of monocytes to CD206^+^ TAMs via PAI-1 and IL-8 production. Moreover, the numbers of IL-8^+^ and PAI-1^+^ cells are significantly associated with the progression-free survival rate. A schematic model for the interactions between OSCC cells and TAM subsets in the TME, established by our present and recent studies in OSCC, is shown in Figure 6. 

Additional research is required to elucidate the function of TAM subsets by cDNA arrays and single-cell RNA sequencing, because TAM-specific markers have not yet been identified. Qian et al. [26] proposed the following six phenotypes of TAMs: activated, immunosuppressive, angiogenic, metastasis-associated, perivascular, and invasive macrophages. These macrophage phenotypes were defined by canonical markers (CD11b, F4/80, and CSF-1R), and the authors found that TAMs show both the phenotype of M2 macrophages, as well as M1-like macrophages. In addition, Komohara et al. [27] suggested that TAMs contain various macrophage phenotypes with a wide range of polarization statuses stimulated by multiple signaling in the TME. A more thorough understanding of the role of TAM subsets and their distribution in OSCC could lead to the identification of a specific marker for TAM subsets and the eventual development of novel pharmacological strategies to interrupt the interaction between OSCC cells and TAM subsets as a further means of inhibiting tumor progression.

**Table 1 ijms-22-09475-t001:** Reported effects of tumor-associated macrophages (TAMs) on oral squamous cell carcinoma (OSCC).

Principal Findings	References
CD206^+^ TAMs might play a key role in OSCC proliferation via EGF production	[23]
CAFs differentiate monocytes to TAMs via the tCXCL12/CXCR4 pathway and aid in the formation of CSC-like cells to enhance OSCC proliferation with reduced apoptosis	[28]
TAM and angiogenesis profiles imply a nonimmunosuppressive mechanism in young and elderly OSCC patients	[29]
PFKFB3 may promote angiogenesis in tumor progression and metastasis by regulating CD163^+^ TAM infiltration in OSCC	[30]
Infiltration by Tregs and M2 TAMs is associated with the progression of premalignant lesions to OSCC	[31]
CD163^+^- and CD8^+^-infiltrating cells influence early and subsequent stages of oral carcinogenesis	[32]
TGF-1β promotes OSCC-associated macrophages to secrete more VEGF via TβRII/Smad3 signaling	[33]
CD163^+^/CD204^+^ TAMs may affect OSCC invasion and metastasis by regulating T cells via IL-10 and PD-L1	[19]
CD163^+^ TAMs promote lymphangiogenesis by expressing VEGF-C, which contributes to regional lymph node metastasis in OSCC	[34]
CAFs shape the OSCC immunosuppressive microenvironment by including the TAM protumoral phenotype	[35]
OSCCs directly suppress antitumor T cell immunity by conditioning TAMs	[36]
CD11b^+^ myeloid cells and CD206^+^ M2 macrophages increase during human OSCC recurrence after radiotherapy	[37]
TAMs promote the EMT of cancer cells, thereby leading to the progression of oral cancer	[38]
MMP and TAM expressions are inversely related in OSCC primary and metastatic regions	[39]
Axl signaling of OSCC is involved in polarizing TAMs toward the M2 phenotype	[40]
CD163^+^ cells may be effective predictors of OSCC prognosis	[41]
CAFs and TAMs expression may help guide treatment decisions to improve survival of OSCC patients	[42]
IL-8 affects the generation of CD163^+^ M2 macrophages in OSCC, which produces immune-suppressive cytokines such as IL-10	[43]
TAMs have a protumor function in OSCC and likely promote tumor progression by activating Gas6/Axl-NF-κB signaling	[44]
TAM markers are associated with CSC markers and OSCC overall survival, suggesting their potential prognostic value in OSCC	[45]
TAMs and angiogenesis affect different histological grades of OSCC	[46]
CAFs and CD163^+^ macrophages may be potential prognostic predictors of OSCC	[47]
Infiltrated TAMs in OSCC have a M2 phenotype and may affect OSCC development and progression	[48]
Increased TAMs are associated with angiogenesis and higher histopathological grades in oral cancer	[49]
The presence of CD163 expression in oral tongue squamous cell carcinomas was associated with worse disease-free survival	[50]

Abbreviations: CAFs, cancer-associated fibroblasts; CSC, cancer stem cell; EGF, epidermal growth factor; EMT, epithelial-to-mesenchymal transition; Gas6, growth arrest-specific gene-6; IL-10, interleukin 10; MMP, matrix metalloproteinase; NF-κB, nuclear factor-kappa B; PD-L1, programmed death-ligand 1; Smad3, mothers against decapentaplegic homolog 3; TGF, transforming growth factor; Tregs, regulatory T cells; VEGF, vascular endothelial growth factor.

## 4. Materials and Methods

### 4.1. Isolation of Peripheral Blood Monocytes

Blood samples were taken from three healthy donors at Kyushu University Hospital. CD14^+^ monocytes were isolated following the previously reported method [21]. Blood samples were drawn into BD Vacutainer cell preparation tubes (BD Biosciences, San Jose, CA, USA), and peripheral blood mono-nuclear cells (PBMCs) were isolated by centrifugation. CD14^+^ monocytes were obtained using the EasySep™ Human Monocyte Isolation Kit (STEMCELL Technologies Inc., Vancouver, BC, Canada). The purity of the isolated cells was validated using flow cytometric analysis (Appendix A)**.**

### 4.2. Co-Culture of Monocytes and OSCC Cell Lines

The HSC-2 cell line was established from metastatic cervical lymph node lesions of OSCC of the floor of the mouth, while SQUU-A and SQUU-B cell lines were established from recurrent tongue cancer of the same patient by orthotopic implantation. Cells were co-cultured in 6-well Transwells with inserts containing 0.4 μm membrane filters (Corning Falcon, Corning, NY, USA). Isolated monocytes (3 × 10^5^) were plated on 6-well plates, and OSCC cells (4 × 10^4^) were plated in the insert. Isolated monocytes and OSCC cells were cultured in RPMI1640 (GIBCO BRL, Grand Island, NY, USA) with 2% FBS (Sigma-Aldrich, St. Louis, MO, USA) in a humidified chamber at 37 °C with 5% CO_2_. The cells were co-cultured for 4 days, and media were refreshed every other day. 

### 4.3. Flow Cytometric Analysis

Cells were washed with Flow Cytometry Staining Buffer (BD Biosciences). After rinsing, the cells were blocked with human FcR blocking reagent (Miltenyi Biotec, Bergisch Gladbach, Germany) for 10 min and incubated at room temperature for 30 min in the dark with PE anti-human CD163 antibodies (Clone GHI/61, IgG1, κ; BioLegend, San Diego, CA, USA), APC anti-human CD204 antibodies (Clone 7C9C20, IgG2a, κ; BioLegend), and FITC anti-human CD206 antibodies (Clone 15-2, IgG1, κ; BioLegend). We used 7-aminoactinomycin D (7-AAD; Biolegend), PE mouse IgG1, κ (BioLegend), APC mouse IgG2a, κ (BioLegend), and FITC mouse IgG1, κ (BioLegend) as isotype control antibodies. Cells were analyzed using a BD FACSVerse™ Flow Cytometer (BD Biosciences), and FlowJo software 10.7.0 (Treestar, Ashland, OR, USA) was used to examine cells and analyze FACS data. 

### 4.4. Cytokine Array

The CMs of HSC-2, SQUU-A, and SQUU-B cells were collected, and soluble proteins were detected using the Proteome Profiler Human XL Cytokine Array Kit (R&D Systems, Minneapolis, MN, USA) in accordance with the manufacturer’s guidelines. For imaging, we used a luminograph.

### 4.5. Enzyme-Linked Immunosorbent Assay (ELISA)

The CM of HSC-2, SQUU-A, and SQUU-B cells were examined using the human IL-8 Quantikine ELISA Kit (R&D Systems) and human SerpinE1 Quantikine ELISA Kit (R&D Systems) in accordance with the manufacturer’s guidelines. RPMI1640 with 2% FBS served as the negative control. IL-8 and SerpinE1 concentrations were measured using the Multiskan FC microplate reader (Thermo Fisher Scientific, Waltham, MA, USA) at a wavelength of 450 nm.

### 4.6. Cell Culture and Stimulation of Human Monocytes In Vitro

Isolated monocytes (3 × 10^5^) were plated on 6-well Transwells. Cells were stimulated using rhIL-8 (2.5 μg/mL; BioLegend) and rhSerpinE1 (2.5 μg/mL; BioLegend) in RPMI1640 with 2% FBS. The concentration of IL-8 and PAI-1 was decided based on the expression of TAM markers with several different numerical concentrations (Appendix A). Monocytes were cultured at 37 °C with 5% CO_2_ for 4 days, and the medium and cytokines were refreshed every other day.

### 4.7. Immunohistochemical Analysis

We obtained surgical specimens from 30 patients with primary tongue OSCC who were treated in the Department of Oral and Maxillofacial Surgery at Kyushu University Hospital from 2005 to 2018. The sections were deparaffinized in xylene and then hydrated by graded series of ethanol. The sections were then incubated with the following primary antibodies at room temperature: mouse anti-CD163 (Clone 10D6, 1:400 dilution; Novocastra, Newcastle, UK), rabbit anti-CD204 (ab217843, 1:200 dilution; Abcam, Tokyo, Japan), rabbit anti-CD206 (ab64693, 1:1000 dilution; Abcam), rabbit anti-IL-8 (ab106350, 1:500 dilution; Abcam), and rabbit anti-PAI-1 (ab66705, 1:1000 dilution; Abcam). Samples were washed with TBST, and 100–400 μL of DAB (Peroxidase Stain DAB Kit; Nacalai Tesque, Kyoto, Japan) was applied as a chromogen to each section. Finally, Mayer’s hemalum solution (1:4 dilution; Merck KGaA, Darmstadt, Germany) was used for counterstaining, and then sections were washed twice for 5 min each in dH_2_O. After dehydration, sections were mounted with coverslips. 

### 4.8. Triple Immunofluorescence Analysis

Immunofluorescence analysis was performed as previously described [21]. The slides were microwaved in AR9 buffer (Opal-4 Color Manual IHC kit; PerkinElmer, Waltham, MA, USA) and cooled for 30 min. Sections were then incubated in Antibody Diluent/Blocking Buffer (Opal-4 Color Manual IHC kit; PerkinElmer) for 10 min at room temperature and then incubated with primary antibodies (listed above). Triple staining was performed with antibodies against IL-8 and SerpinE1 overnight at 4 °C and antibodies against CD206 for 2 h at room temperature. Samples were washed three times in TBST for 2 min each and then incubated in Polymer HRP (Ms + Rb) (Opal-4 Color Manual IHC kit; PerkinElmer). Samples were rinsed and then washed three times in TBST for 2 min each and then incubated in Opal Fluorophore working solution (TSA Plus System; PerkinElmer) for 10 min at room temperature. Samples were rinsed in TBST, microwaved in AR9 buffer, and then mounted with VECTASHIELD Mounting Medium For Fluorescence with DAPI (Vector Laboratories, Burlingame, CA, USA). Photomicrographs were obtained using a light microscope with a digital camera (BZ-X800 series; Keyence, Tokyo, Japan). The ratios of triple IL-8-, SerpinE1-, and CD206-positive cells in immune cells and tumor areas were quantified using the BZ-H4C and BZ-H4CM analytic applications.

### 4.9. Statistical Analysis 

JMP software version 15 (SAS Institute, Cary, NC, USA) was used to perform statistical analyses. The significant differences between each group were assessed by the Kruskal–Wallis test followed by post hoc Steel’s test. Correlations were measured using Pearson’s correlation coefficient. In all analyses, *p* values ≤ 0.05 were considered statistically significant. 

## Figures and Tables

**Figure 1 ijms-22-09475-f001:**
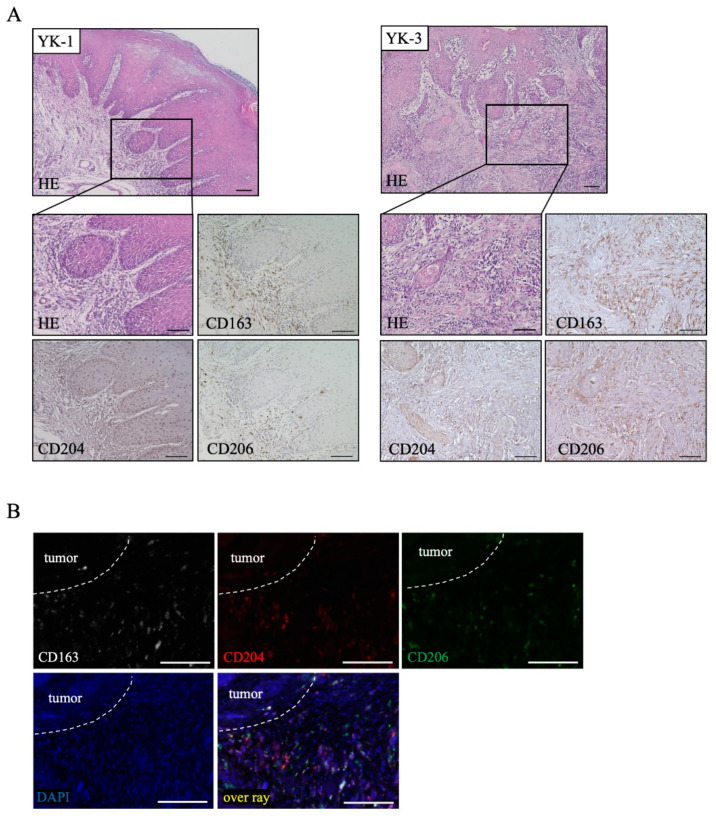
Distribution of tumor-associated macrophages (TAMs) in oral squamous cell carcinoma (OSCC). (**A**) Immunohistochemical staining of OSCC sections with H&E and CD163, CD204, and CD206 antibodies (brown). Scale bars: 100 μm. (**B**) Triple immunofluorescence staining was performed for CD163 (white), CD204 (red), and CD206 expressions (green); nuclei were stained with DAPI (blue). Scale bars: 100 μm. YK, Yamamoto–Kohama criteria.

**Figure 2 ijms-22-09475-f002:**
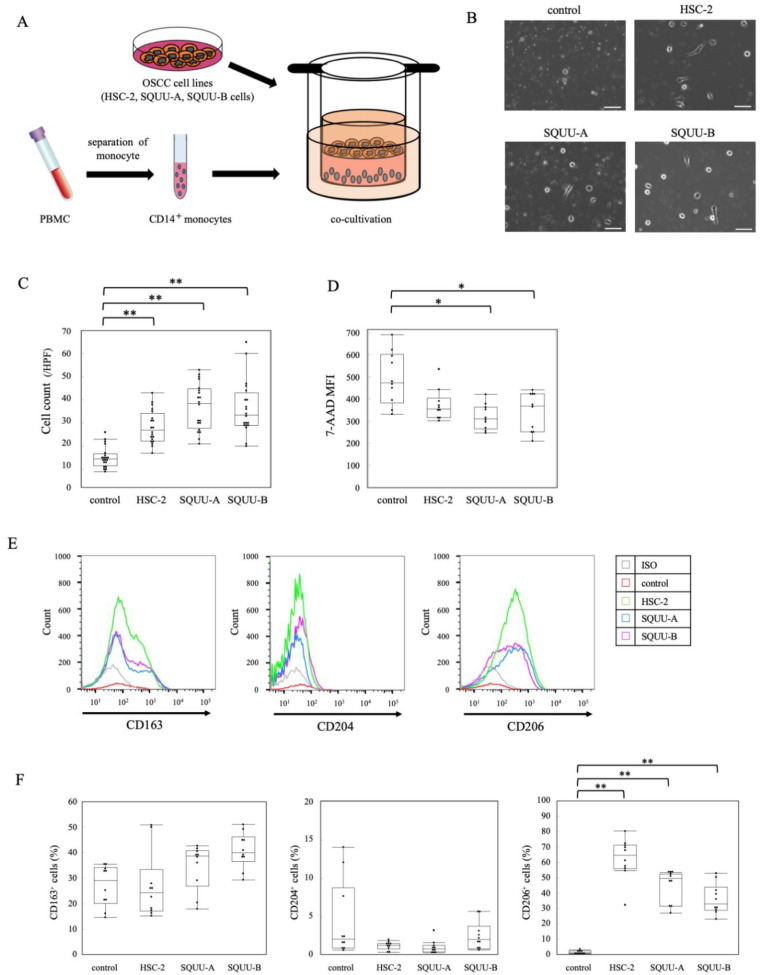
Effects on the differentiation of monocytes to TAM subsets by co-cultivation with OSCC cell lines. (**A**) Schematic illustration of the co-cultivation of isolated CD14^+^ monocytes and OSCC cell lines (HSC-2, SQUU-A, and SQUU-B cells) for 4 days. (**B**) Representative images of CD14^+^ monocytes alone (control) or in co-culture with OSCC cell lines. (**C**) The mean number of CD14^+^ monocytes (*n* = 15 for each group) were counted in a high power field (HPF), from 5 different areas. (**D**) The mean fluorescence intensity (MFI) of 7-AAD^+^ CD14^+^ monocytes co-cultured with OSCC cell lines (*n* = 10 for each group) were analyzed by flow cytometry. The detection (**E**) and positive rate (**F**) of CD163^+^, CD204^+^, and CD206^+^ cells co-cultured with OSCC cell lines for 4 days (*n* = 10 for each group), as determined by flow cytometric analysis. * *p* < 0.05; ** *p* < 0.01 by Kruskal–Wallis test followed by post hoc Steel’s test.

**Figure 3 ijms-22-09475-f003:**
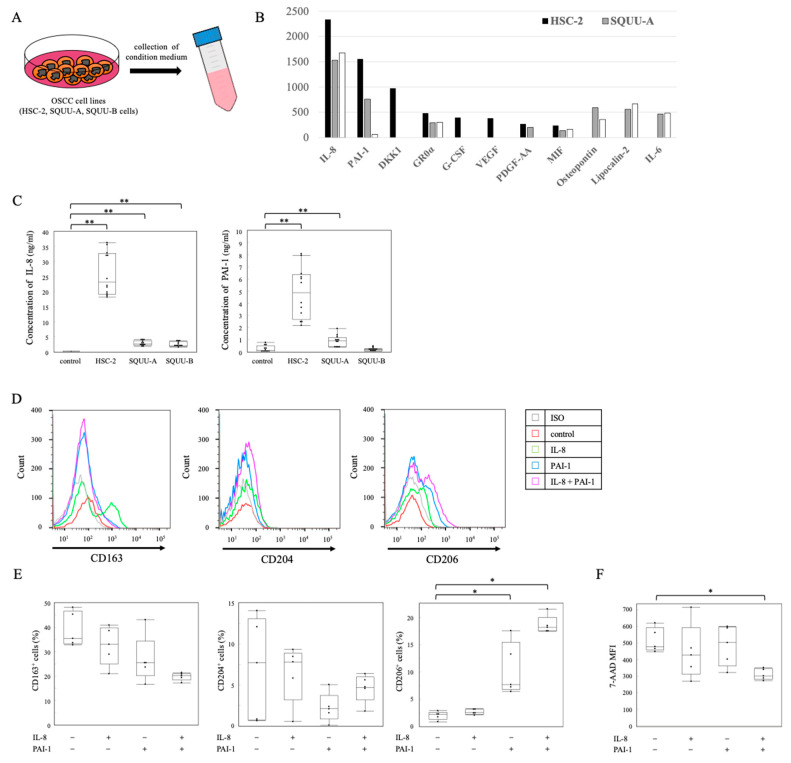
Differentiation of monocytes to TAM subsets by OSCC-produced cytokines. (**A**) Schematic illustration of collection of conditioned medium (CM) from OSCC cell lines cultured for 48 h. (**B**) The expression of cytokines produced by OSCC cell lines was analyzed by cytokine array. (**C**) The concentrations of interleukin-8 (IL-8) and plasminogen activator inhibitor-1 (PAI-1) in the CM of OSCC cell lines (*n* = 10 for each group) were measured by enzyme-linked immunosorbent assay. The detection (**D**) and positive rate (**E**) of CD163^+^, CD204^+^, and CD206^+^ cells co-cultured with IL-8 and/or PAI-1 for 4 days (*n* = 5 for each group), as determined by flow cytometric analysis. (**F**) The MFI of 7-AAD^+^ CD14^+^ monocytes co-cultured with OSCC cell lines (*n* = 5 for each group) was analyzed by flow cytometry. Bars show the mean ± SD. * *p* < 0.05; ** *p* < 0.01 by Kruskal–Wallis test followed by post hoc Steel’s test.

**Figure 4 ijms-22-09475-f004:**
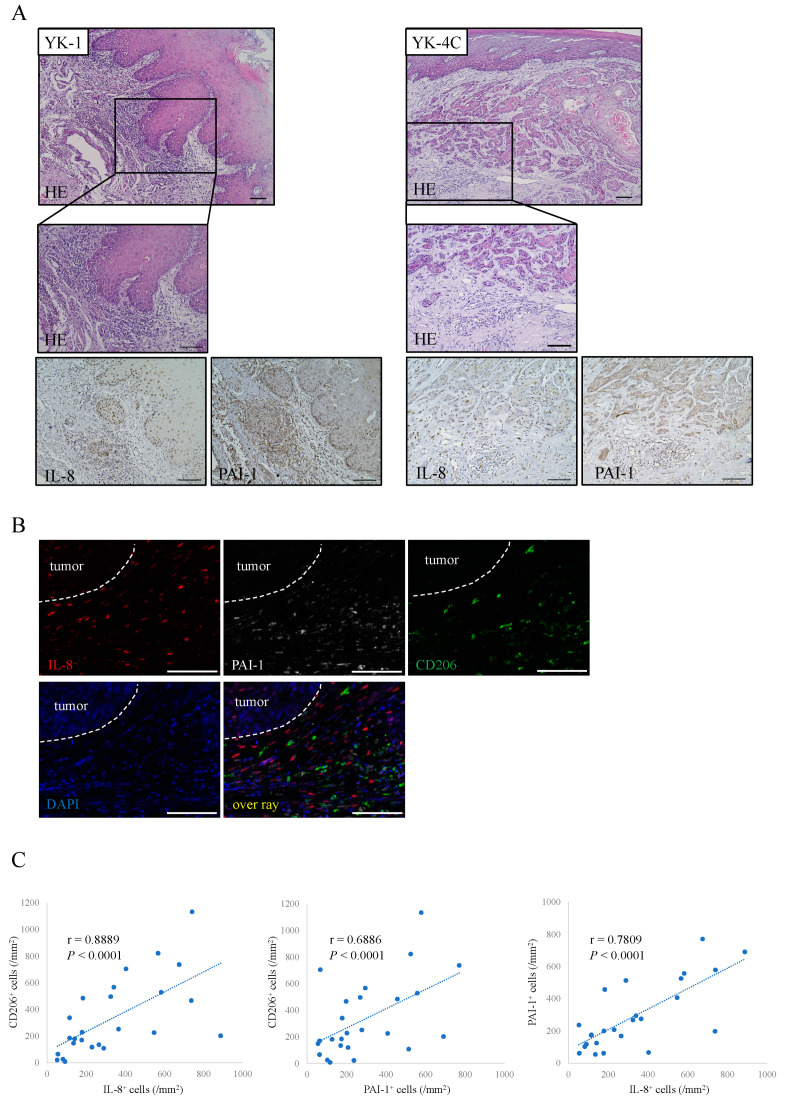
Distribution of IL-8 and PAI-1 in OSCC. (**A**) Immunohistochemical staining of OSCC sections with H&E and IL-8 and PAI-1 antibodies (brown). Scale bars: 100 μm. (**B**) Triple immunofluorescence staining was performed with antibodies against IL-8 (red), PAI-1 (white), and CD206 (green); nuclei were stained with DAPI (blue). Scale bars: 100 μm. (**C**) Correlations among the numbers of CD206^+^, IL-8^+^, and PAI-1^+^ cells in 18 OSCC patients. Statistically significant differences between groups were determined by Pearson’s correlation coefficient.

**Figure 5 ijms-22-09475-f005:**
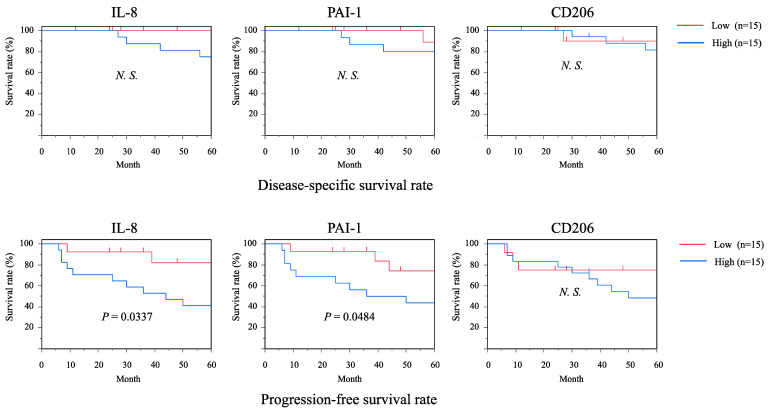
Survival curves according to the expressions of IL-8, PAI-1, and CD206 in OSCC tissues. Survival rates were calculated by the Kaplan–Meier method with a high versus low expression of IL-8^+^, PAI-1^+^, and CD206^+^ cells. The classifications are described in the Results section. Statistically significant differences between groups were determined by the log-rank test.

**Figure 6 ijms-22-09475-f006:**
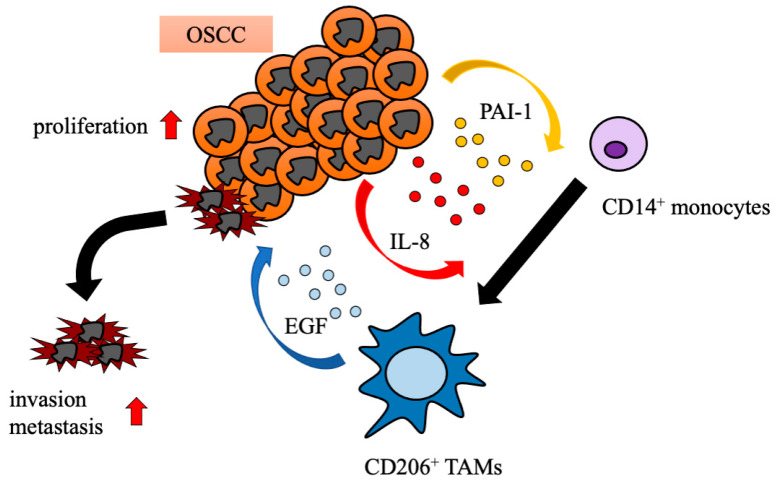
Schematic model of the interaction between OSCC and TAM subsets. OSCC cells enhance CD14^+^ monocytes to differentiate into CD206^+^ TAMs via PAI-1 and IL-8 production, and CD206^+^ TAMs shape the tumor microenvironment toward tumor progression, including increased proliferation, invasion, and metastasis via epidermal growth factor (EGF) production.

## Data Availability

The datasets generated during and/or analyzed during the current study are available from the corresponding author on reasonable request.

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
