# Peer review of "Oral Squamous Cell Carcinoma Contributes to Differentiation of Monocyte-Derived Tumor-Associated Macrophages via PAI-1 and IL-8 Production"

_ijms, 2021, doi:10.3390/ijms22179475_

Round 1

Reviewer 1 Report

General comments

In this study, Kai et al examined the effect of oral squamous cell carcinoma on the differentiation of

monocyte-derived tumor-associated macrophages using expression levels of cell surface markers including CD163, CD204, and CD206. They showed that the expression of CD206 on CD14+ cells was significantly increased after co-culture with OSCC cell lines. They also showed high concentrations of plasminogen activator inhibitor-1 (PAI-1) and interleukin-8 (IL-8) in the conditioned medium of OSCC cell lines. Although interesting, the results do not necessarily reflect the clinical setting due to use of the monocytes from healthy donors. Therefore, the author should generate more data to reach significant in the study.

  1. The most important point is to use primary tumor associated macrophages and cancer cells from OSCC patients. The author should utilize such TAMs and/or cancer cells to evaluate the findings in the study.
  2. 2C: The author should show the representative dot plots of 7-AAD stained macrophages. Why the authors show the cell viabilities with mean fluorescence intensity? The dead cells would be clearly distinguished from living cells.
  3. 2D: The authors show the result of cell count. What is the unit of the cell count? Is this percentage? Please specify the unit of that.
  4. 2F: The author shows the percentages of cell surface markers. The reviewer wonders why the percentages become 100% in the case of HSC-2 but not in that of control? How did the authors calculate the percentages? Does it mean that CD163-/CD204-/CD206- cells are dominant in the control group? Please explain the result more accurately in the Results section.
  5. 3D: It is quite difficult to distinguish the colors of histogram among the groups. The author should clearly show the histogram and legend with different colors.
  6. 3E: The result of CD163+ cells (%) seems inconsistent with the result in Supplementary Figure 2. In Fig. 3E, the percentages of CD163+ cells seem decreases after IL-8 and/or PAI-1 stimulation. However, they seem an increase after PAI-1 stimulation in the Supplementary Figure 2. Also, the CD206+ cells rates (%) in the Fig. 3E are different from those in the Supplementary Figure 2. Please explain the discrepancies.
  7. Fig 4C: The authors showed the correlation between the number of CD206+ and IL-8+ cells, CD206+ and PAI-1+ cells, and PAI-1+ and IL-8+ cells Does the number of IL-8+ and PAI-1+ cells mean the number of IL-8+ and/or PAI-1+ tumor cells? The authors should clarify this. Since macrophages would be typically positive for IL-8 or PAI-1 expression, the author should analyze the correlation between CD206+ macrophages and IL-8+ or PAI-1+ tumor cells. This would be an impact on the finding in this study.
  8. Please specify the Supplementary Figure 2 in the manuscript. Also, please add the error bars and specify the number of experiments in the Supplementary Figure 2.
  9. The authors preciously reported that TAM-derived EGF promote the OSCC proliferation. How about the effect of EGF on the expression levels of IL-8 and PAI-1 in OSCC cells? Would it be possible that anti-EGFR treatment decrease the percentage of CD206+ cells, IL-8 and PAI-1 expression?

Author Response

Response to Reviewer 1 Comments

In this study, Kai et al examined the effect of oral squamous cell carcinoma on the differentiation of monocyte-derived tumor-associated macrophages using expression levels of cell surface markers including CD163, CD204, and CD206. They showed that the expression of CD206 on CD14+ cells was significantly increased after co-culture with OSCC cell lines. They also showed high concentrations of plasminogen activator inhibitor-1 (PAI-1) and interleukin-8 (IL-8) in the conditioned medium of OSCC cell lines. Although interesting, the results do not necessarily reflect the clinical setting due to use of the monocytes from healthy donors. Therefore, the author should generate more data to reach significant in the study.

Point 1: The most important point is to use primary tumor associated macrophages and cancer cells from OSCC patients. The author should utilize such TAMs and/or cancer cells to evaluate the findings in the study. 

Response 1: As you pointed, in vitro study using primary tumor will reflect the clinical setting more accurately. However, it is quite difficult to extract the number of monocytes enough for our in vitro examination from OSCC tissue. Because it takes a great deal of tissue volume for that and indicating to interfere with pathological test of OSCC section. Therefore, we examine the association of CD206, PAI-1, and IL-8 expression with clinical findings of OSCC patients and added these results as Figure 5.

Point 2: Fig. 2C: The author should show the representative dot plots of 7-AAD stained macrophages. Why the authors show the cell viabilities with mean fluorescence intensity? The dead cells would be clearly distinguished from living cells.

Response 2: We added the representative dot plots of 7-AAD stained macrophages (CD163+, CD204+, and CD206+cells) as Supplementary Figure 1. In apoptotic cells, the membrane phospholipid phosphatidylserine is translocated from the inner leaflet to the outer leaflet of the plasma membrane. 7-AAD is a standard viability probe for distinguishing viable from nonviable cells. Viable cells with intact membranes exclude 7-AAD, whereas damaged membranes are permeable to 7-AAD. Thus, our current and previous studies adopted mean fluorescence intensity (MFI) instead of positive rate for 7-AAD staining.

Point 3: Fig. 2D: The authors show the result of cell count. What is the unit of the cell count? Is this percentage? Please specify the unit of that.

Response 3: “The unit of the cell count” means the mean number of monocytes in high power field, from 5 different areas. We described the meaning of “the unit of the cell count” in figure legend. Since Fig. 2D is associated with Fig. 2B, we changed the order of Fig. 2D to Fig. 2C. 

Point 4: Fig. 2F: The author shows the percentages of cell surface markers. The reviewer wonders why the percentages become 100% in the case of HSC-2 but not in that of control? How did the authors calculate the percentages? Does it mean that CD163-/CD204-/CD206- cells are dominant in the control group? Please explain the result more accurately in the Results section.

Response 4: We calculated the positive rate of each TAM subset (CD163+, CD204+, or CD206+ cells). For example, CD163+ cells include CD163+CD204-CD206-, CD163+CD204+CD206-, CD163+CD204-CD206+, and CD163+CD204+CD206+ cells. Therefore, the total percentage in each group (control, HSC-2, SQUAA) does not always become 100%. We described this analysis method in figure legend.

Point 5: Fig. 3D: It is quite difficult to distinguish the colors of histogram among the groups. The author should clearly show the histogram and legend with different colors.

Response 5: We revised Fig. 3D to become able to distinguish the colors of histogram among the groups.

Point 6: Fig. 3E: The result of CD163+ cells (%) seems inconsistent with the result in Supplementary Figure 2. In Fig. 3E, the percentages of CD163+ cells seem decreases after IL-8 and/or PAI-1 stimulation. However, they seem an increase after PAI-1 stimulation in the Supplementary Figure 2. Also, the CD206+ cells rates (%) in the Fig. 3E are different from those in the Supplementary Figure 2. Please explain the discrepancies.

Response 6: We revised the result in Supplementary Figure 2 (revise version: Supplementary Figure 3) to consistent with the result of Fig. 3E.

Point 7: Fig. 4C: The authors showed the correlation between the number of CD206+ and IL-8+ cells, CD206+ and PAI-1+ cells, and PAI-1+ and IL-8+ cells Does the number of IL-8+ and PAI-1+ cells mean the number of IL-8+ and/or PAI-1+ tumor cells? The authors should clarify this. Since macrophages would be typically positive for IL-8 or PAI-1 expression, the author should analyze the correlation between CD206+ macrophages and IL-8+ or PAI-1+ tumor cells. This would be an impact on the finding in this study.

Response 7: As you pointed, PAI and IL-8 are reported to be produce by tumor cells as well as macrophages. However, we have confirmed the co-expression among CD206, IL-8, and PAI (Figure 4B), and it was actually different from its distribution. These results suggest that the production of PAI and IL-8 by CD206+ macrophages was remarkably less than that in tumor cells. Therefore, we simply analyzed the correlation between CD206, IL-8, and PAI expression.

Point 8: Please specify the Supplementary Figure 2 in the manuscript. Also, please add the error bars and specify the number of experiments in the Supplementary Figure 2.

Response 8: We described specify the Supplementary Figure 2 (revise version: Supplementary Figure 3) in the Materials and Methods section. To appropriately correct the concentration of IL-8 and PAI, we preformed the stimulation tests in a dose-dependent manner in only one experiment. Thus, the error bars were not existed.

Point 9: The authors preciously reported that TAM-derived EGF promote the OSCC proliferation. How about the effect of EGF on the expression levels of IL-8 and PAI-1 in OSCC cells? Would it be possible that anti-EGFR treatment decrease the percentage of CD206+ cells, IL-8 and PAI-1 expression?

Response 9: Admittedly, previous studies demonstrated that EGFR signaling induced IL-8 and PAI-1 production in pulmonary epithelial cells or keratinocytes. (Toxicol Sci. 2019169(2):534-542, J Invest Dermatol. 2010;130(9):2179-90). Although we thus speculated that anti-EGFR treatment might decrease the percentage of CD206+ cells, IL-8 and PAI-1 expression in OSCC tissue, it takes a great deal of time to clarify the effect of anti-EGFR treatment (it cannot be finished at least by 10 days). Instead, we described the possible effect of EGFR on IL-8 and PAI-1 expression in the Discussion section and added the references (ref No. 22, 23).

Reviewer 2 Report

Kai et al., identified that high concentrations of PAI-1 and IL-8 secreted from OSCC cell lines stimulated CD14+ cells to express CD206, and the numbers of CD206+, PAI-1+, and IL-8+ cells were positively correlated in OSCC biopsies. They suggest that PAI-1 and IL-8 produced by OSCC induce the differentiation of monocytes to CD206+ TAMs.

These findings are very preliminary and lack of functional studies for the polarized M2 macrophages such as invasion, migration and phagocytotic abilities. Also, the knocked down effects of PAI-1 and IL-8 in OSCC on the M2 macrophage plasticity and function should be included in the paper. Moreover, the xenograft mice study should be conducted after silencing of PAI-1 and IL-8 in OSCC cells. Furthermore, the infiltration of M2 macrophages need to be confirmed in the tumors deficient of PAI-1 and IL-8.

Author Response

Response to Reviewer 2 Comments

Kai et al., identified that high concentrations of PAI-1 and IL-8 secreted from OSCC cell lines stimulated CD14+ cells to express CD206, and the numbers of CD206+, PAI-1+, and IL-8+ cells were positively correlated in OSCC biopsies. They suggest that PAI-1 and IL-8 produced by OSCC induce the differentiation of monocytes to CD206+ TAMs.

Point 1: These findings are very preliminary and lack of functional studies for the polarized M2 macrophages such as invasion, migration and phagocytotic abilities. Also, the knocked down effects of PAI-1 and IL-8 in OSCC on the M2 macrophage plasticity and function should be included in the paper. Moreover, the xenograft mice study should be conducted after silencing of PAI-1 and IL-8 in OSCC cells. Furthermore, the infiltration of M2 macrophages need to be confirmed in the tumors deficient of PAI-1 and IL-8.

 Response 1: With respect to invasion and migration abilities of polarized M2 macrophages, we have previously demonstrated the effect of proliferation and invasion in OSCC cell lines by co-culture with CM of TAM subsets (CD163, CD204, and CD206+ cells) analyzed by cell proliferation and invasion assays (ref 21). We described these results and comments in the Discussion section.

As you pointed, knocked down and xenograft mice studies will clarify the function of polarized M2 macrophages and OSCC cells. However, it takes a great deal of time for that (it cannot be finished at least by 10 days). Instead, we added the association of CD206, PAI-1, and IL-8 expression with clinical findings of OSCC patients as Figure 5.

Reviewer 3 Report

In this study the authors investigated the influence of cancer cells on monocyte differentiation to tumor-associated macrophages in oral cancer. The results were interesting, showing that the expression of CD206 on CD14 cells was increased after co-culture with oral cancer cell lines; on the contrary, the expressions of CD163 and CD204 on CD14 cells showed no change. Moreover, PAI-1 and IL-8 stimulated CD14 cells to express CD206, and there were positive correlations among the numbers of cells expressing CD206, PAI-1, and IL8 in oral cancer sections.

The techniques utilized were appropriate and described with plenty details. This is a well-designed study with rigorous methods. The discussion is well-balanced, and the statements are supported by the data. This paper is very interesting and on a timely subject in view of increasing interest about immune-oncology.

In the Discussion section, the authors reported the effects of TAMs on oral cancer. I recommend updating the literature through read and discuss the recent work of Agarbati et al. (for your convenience, DOI: 10.1097/PAI.0000000000000867).

Author Response

Response to Reviewer 3 Comments

In this study the authors investigated the influence of cancer cells on monocyte differentiation to tumor-associated macrophages in oral cancer. The results were interesting, showing that the expression of CD206 on CD14 cells was increased after co-culture with oral cancer cell lines; on the contrary, the expressions of CD163 and CD204 on CD14 cells showed no change. Moreover, PAI-1 and IL-8 stimulated CD14 cells to express CD206, and there were positive correlations among the numbers of cells expressing CD206, PAI-1, and IL8 in oral cancer sections.

The techniques utilized were appropriate and described with plenty details. This is a well-designed study with rigorous methods. The discussion is well-balanced, and the statements are supported by the data. This paper is very interesting and on a timely subject in view of increasing interest about immune-oncology.

Point 1: In the Discussion section, the authors reported the effects of TAMs on oral cancer. I recommend updating the literature through read and discuss the recent work of Agarbati et al. (for your convenience, DOI: 10.1097/PAI.0000000000000867).

 Response 1: Thank you for your comment. We described the effects of TAMs on oral cancer reported by Agarbati et al in Table 1 and added the reference (No. 48).

Round 2

Reviewer 1 Report

The authors properly replied to the reviewer's comments. The revised form would be acceptable.

In the revised manuscript, the reviewer found that they miss the molecular names in the newly-added Figure 5 (They might be IL-8, PAI-1, and CD206). They should add the names as headings in each graph.

Author Response

Response to Reviewer 1 Comments

The authors properly replied to the reviewer's comments. The revised form would be acceptable.

Point 1: In the revised manuscript, the reviewer found that they miss the molecular names in the newly-added Figure 5 (They might be IL-8, PAI-1, and CD206). They should add the names as headings in each graph. 

 Response 1: As you pointed, we added molecular names (IL-8, PAI-1, and CD206) in the Figure 5.

Reviewer 2 Report

The clinical evidence should be also confirmed by the analysis of commercial cDNA array.

Author Response

Response to Reviewer 2 Comments

Point 1: The clinical evidence should be also confirmed by the analysis of commercial cDNA array.

Response 1: As you pointed, he analysis of commercial cDNA array will clarify he association of CD206, PAI-1, and IL-8 expression with clinical findings of OSCC patients. However, it takes a great deal of time for that (it cannot be finished at least by 24 hours). Instead, we described the necessity of cDNA array and single-cell RNA sequencing to elucidate the clinical evidence in the Discussion section.